# Mental Fatigue, Activities of Daily Living, Sick Leave and Functional Status among Patients with Long COVID: A Cross-Sectional Study

**DOI:** 10.3390/ijerph192214739

**Published:** 2022-11-09

**Authors:** Trine Brøns Nielsen, Steffen Leth, Mathilde Pedersen, Helle Dahl Harbo, Claus Vinther Nielsen, Cecilia Hee Laursen, Berit Schiøttz-Christensen, Lisa Gregersen Oestergaard

**Affiliations:** 1DEFACTUM, Central Denmark Region, 8000 Aarhus, Denmark; 2Department of Physiotherapy and Occupational Therapy, Aarhus University Hospital, 8200 Aarhus, Denmark; 3Department of Infectious Diseases, Aarhus University Hospital, 8200 Aarhus, Denmark; 4Department of Infectious Diseases and Internal Medicine, Gødstrup Regional Hospital, 7400 Herning, Denmark; 5Department of Clinical Medicine, Aarhus University, 8200 Aarhus, Denmark; 6AmbuFlex—Center for Patient-Reported Outcomes, Gødstrup Regional Hospital, 7400 Herning, Denmark; 7Department of Clinical Social Medicine and Rehabilitation, Gødstrup Regional Hospital, 7400 Herning, Denmark; 8Department of Public Health, Aarhus University, 8000 Aarhus, Denmark; 9Research Unit of General Practice, Department of Public Health, University of Southern Denmark, 5000 Odense, Denmark

**Keywords:** long COVID, persistent COVID-19 symptoms, activities of daily living, mental fatigue, functioning, rehabilitation, work ability

## Abstract

Studies suggest that persistent symptoms after COVID-19 (long COVID) influence functioning and activities of daily living (ADL). However, it is still uncertain how and to what extent. This study aimed to describe patient-reported mental fatigue, ADL problems, ADL ability, sick leave and functional status among patients with long COVID. In this cross-sectional study, 448 patients, ≥18 years old and referred to occupational therapy at a Danish Post-COVID-19 Clinic, were included. Mental fatigue was measured by the Mental Fatigue Scale, ADL problems and ability were measured by the Canadian Occupational Performance Measure, sick leave was self-reported and functional status was evaluated by the Post-COVID-19 Functional Status Scale. Mean age was 46.8 years, 73% of the patients were female, and 75% suffered from moderate to severe mental fatigue. The majority reported difficulties performing productive and leisure activities. The median performance and satisfaction scores were 4.8 and 3, respectively. In total, 56% of the patients were on sick leave, and 94% were referred to rehabilitation. A decrease in functional status was found between pre-COVID-19 and assessment. Conclusively, the patients were highly affected in their everyday life and had distinct rehabilitation needs. Future research is needed to address causalities and rehabilitation for this patient group.

## 1. Introduction

With more than 584 million confirmed cases and 6 million deaths reported to the WHO [1], the COVID-19 pandemic has affected millions of lives and a multitude of societies worldwide in the past few years. The majority of people infected with SARS-CoV-2 virus causing COVID-19 recover without treatment and return to normal life [2]. Although most symptoms resolve after a few weeks, some individuals experience symptoms that extend beyond 12 weeks. In line with the definition by the WHO, these symptoms are referred to as long COVID [3].

In a follow-up study of hospitalised patients, the prevalence of long COVID has been reported to reach 76% (1265 of 1655) at six months of follow-up [4], dropping to 55% (650 of 1190) after two years [5]. The reported prevalence of long COVID for non-hospitalised patients varies and has typically been studied in selected populations [6]. The WHO estimates that 10% of people with COVID-19 will experience long COVID and argues that the condition is independent of the severity of the initial infection [7]. Even so, with the emergence of new virus variants, such as Omicron, the current prevalence of long COVID might differ from earlier estimates. In a recent observational case-control study completed in the United Kingdom, the prevalence of long COVID decreased to 4.5% among Omicron cases from 10.8% among Delta cases, but following the high number of people infected with Omicron, the incidence of long COVID is expected to rise [8].

Long COVID may affect multiple organ systems, causing a wide range of symptoms such as generalised pain, fatigue, muscle exhaustion, decreased lung function, cough, affected sense of smell and taste, depression, anxiety and cognitive dysfunction [7,9,10,11,12,13,14]. Fatigue, cognitive dysfunction and dyspnoea are the most prevalent reported symptoms [10,11,12,13]. Furthermore, female sex is generally associated with a higher risk of persistent symptoms [15,16,17].

Only a few studies have investigated the impact of long COVID on activities of daily living (ADL) [18,19,20], and here especially, fatigue and cognitive dysfunction were found to influence ADL and work ability [9,19,20,21]. Likewise, long COVID is described to affect core identity activities such as being a parent, a caregiver or an employee [20]. Studies suggest that long COVID may result in full or part time sick leave and reduce the ability to handle regular work tasks in many months after the acute phase of COVID-19 [9,18,22].

Although the results of previous studies suggest that long COVID impacts functional status, leads to long-term sick leave and influences everyday life of the patients, it remains uncertain how and to which extent. Thus, the aim was to describe patient-reported mental fatigue, ADL problems, ADL ability, sick leave and functional status among patients with long COVID.

## 2. Materials and Methods

The study was a descriptive cross-sectional study recruiting patients from 15 July 2020 to 1 February 2022. The cohort inclusion criterion was referral to clinical evaluation by occupational therapists from the outpatient Post-COVID-19 Clinic due to cognitive deficits and fatigue following long COVID. Patients ≥18 years old could be referred to the Post-COVID-19 Clinic if they had been suffering from continuous or newly developed COVID-19-related symptoms from minimum two organ systems and for a minimum of three months after the acute phase of COVID-19. A positive polymerase chain reaction (PCR) or antigen test for infection with SARS-CoV-2 was required. In a few cases with no test confirming acute infection, the doctors at the clinic provided a symptom-based diagnosis of COVID-19 and long COVID from the case history with the patient and the medical record. The patients were excluded from the study if they could not complete any of the instruments due to language barriers or did not speak or understand Danish well enough to comprehend the written and oral information about the purpose of the project and their rights of participation.

Data collection was based on a combination of patient-reported outcomes and evaluation of functional status. Included assessment tools were part of common practise at the Post-COVID-19 Clinic at a Danish university hospital to which patients were referred. Sociodemographic data (gender, age and self-reported living status, education, work status and occupation, main symptoms and time since infection) were extracted through an electronic questionnaire and from hospital medical records. Information on sick leave was obtained through the electronic questionnaire. The occupational therapists employed at the Post-COVID-19 Clinic performed the data collection. The following instruments were used to address the aim of the study:The Mental Fatigue Scale was used to measure the degree of mental fatigue;The Canadian Occupational Performance Measure was used to examine the impact of long COVID on ADL problems and ADL ability;The Post-COVID-19 Functional Status Scale was used to evaluate the consequences of COVID-19 and its impact on functional status.

Study data were collected and managed using the Research Electronic Data Capture (REDCap), which is a secure electronic web-based software platform designed to support data capture for research studies [23,24]. Data were exported to STATA 17 for analysis [25].

### 2.1. Mental Fatigue Scale

The Mental Fatigue Scale is a questionnaire and contains of 15 questions covering affective, cognitive and sensory symptoms along with duration of sleep and daytime variation in symptom severity [26]. For each question, examples of common activities and four related response alternatives are presented with a score ranging from 0 (normal function) to 3 (maximal symptom). It is possible to rate in between the statements [26,27]. A score of 10.5 has been suggested as cut-off [26]. The scale has been evaluated and is found valid for detection of mental fatigue among persons with different diagnoses compared to controls [28]. A high internal consistency among items has been identified [26]. The Danish translation of the questionnaire was used. To our knowledge, the Danish version has not yet been validated. A link to an electronic version of the Mental Fatigue Scale was sent to the patient prior to his or her visit at the clinic. If the patient had not completed the Mental Fatigue Scale beforehand, a paper version was completed immediately before or during the appointment.

### 2.2. Canadian Occupational Performance Measure

The Canadian Occupational Performance Measure (COPM) is a semi-structured interview through which the occupational therapist and the patient jointly identify the patient’s problems in performing ADL within the areas of self-care, productivity and leisure. The importance of the identified ADL problems is rated by the patient on a ten-point visual rating scale (VRS) ranging from 1 (not important) to 10 (extremely important). Subsequently, the patient prioritises up to five activity problems of high relevance and rates performance and satisfaction with the performance of the prioritised activities. The VRS for level of performance ranges from 1 (not able to do it at all) to 10 (able to do it extremely well); the VRS for satisfaction of performance ranges from 1 (not satisfied at all) to 10 (totally satisfied) [29,30,31]. The Danish version of the 5th edition of the COPM was used in the study. Both face and content validity of the Danish version are adequate [32], and the Danish version has moderate reliability regardless of rater experiences, sites or client diagnosis [31]. The activity problems were categorised using the Taxonomic Code for Occupational Performance [32]. As some activity problems may be categorised under more than one subcategory, a list with examples of expected common activity problems under each subcategory was composed by the research group to increase the consistency of the categorisation.

### 2.3. Post-COVID-19 Functional Status Scale

The Post-COVID-19 Functional Status Scale is an ordinal scale with six steps covering grade 0 (no functional limitations), grade 1 (negligible functional limitations), grade 2 (slight functional limitations), grade 3 (moderate functional limitations), grade 4 (severe functional limitations and grade 5 (death) [33,34]. The Danish version of the scale was used [35]. The construct validity of the Post-COVID-19 Functional Status Scale used for adults with COVID-19 is acceptable for the English version [34] and adequate for the Spanish version [36], but no studies have evaluated the validity or the reliability of the Danish version. The instrument was completed during the appointment with the occupational therapist. A current and a pre-COVID-19 infection Post-COVID-19 Functional Status Scale score were completed in order to assess any change in functional status.

### 2.4. Statistical Analysis

Descriptive statistical analyses were conducted using STATA 17. Categorical data were reported as frequencies (*n*) and relative frequencies (%) and ordinal scale variables as median and percentiles (25th; 75th). Normally distributed continuous variables were described by means and standard deviation (SD), and median and percentiles (25th; 75th) were used for non-normally distributed continuous data. The number of patients included in each analysis was reported.

### 2.5. Ethical Statement

All patients meeting the criteria for inclusion received oral and written information regarding the purpose of the study, their rights of withdrawal and confidentiality from the occupational therapist responsible for their clinical evaluation. Written informed consent was obtained from each patient participating in the study. The study was conducted in accordance with the Declaration of Helsinki and approved by the Danish Data Protection Agency (R. no. 1-16-02-655-20). Approval included permission to obtain data from patient-reported questionnaires, clinical examination and medical records. The Danish Ethical Committee for the Central Denmark Region required no registration (R. no. 1-10-72-1-21).

## 3. Results

As shown in Figure 1, 604 patients were assessed at the Post-COVID-19 Clinic in the inclusion period. Among these patients, 562 were referred to the occupational therapy, of which 477 patients were found eligible for participation. In total, 448 patients were included in the study.

### 3.1. Sociodemographic Data

Table 1 shows the sociodemographic characteristics of the study population. The mean age was 46.8 years and 73% were females. The median number of days from SARS-CoV-2 diagnosis to assessment by the occupational therapist was 246 days. Half of the patients (51%) had completed a medium- or long-cycle higher education. The majority of the population (90%) were either employed (full-time, part-time or under subsidy programmes) or enrolled in education before their COVID-19 diagnosis. Additionally, 5% of the patients were unemployed and 6% were on retirement. At the time of the assessment, 56% of the patients employed prior to infection were sick listed, almost all of whom (96%) were on sick leave due to their long COVID diagnosis. Just under three quarters (74%) of the patients were living with a spouse or partner, and almost half (48%) had children living at home.

The patients were asked to report their worst symptom (Table 1). Two patients did not report, and 32 responses were left out because the patients had reported more than one symptom when completing the questionnaire in paper despite instructions to provide only one answer. Thus, we included answers from 414 of the patients. Mental fatigue was reported as the worst symptom by 71% of the patients. Among the 20 (4.8%) patients who reported “other” as their worst symptom, 15 described headache as their worst symptom. Based on the assessment by the occupational therapist, 94% of the patients were referred to rehabilitation in their home municipality. The rest of the cohort did not want a rehabilitation plan or decided, in collaboration with the occupational therapist, that they could handle their long COVID symptoms based on the advice and guidance they received during their assessment.

### 3.2. Mental Fatigue Scale

In total, 447 (99.8%) patients completed the Mental Fatigue Scale (Table 2). The mean score was 18.6, and 75% of the patients were categorised with moderate to severe mental fatigue. Three of the patients did not provide an answer to 24-h variance, but among the 444 patients who did, more than half of them (63%) reported having noticed a 24-h variance in the degree of their symptoms; among these patients, 76% of the patients felt best in the morning and 80% felt worst either in the afternoon or evening.

### 3.3. Canadian Occupational Performance Measure

In total, 430 (96%) patients completed the COPM, of whom 15 (3%) did not prioritise their activity problems. Reasons for not completing the COPM were typically time limitations of the assessment, the burden of their symptoms or other ethical reasons. A total of 60 activity problems from 32 patients were left out as they could not be categorised as either an activity or occupation according to the Taxonomic Code for Occupational Performance or because they were formulated in a way that made them difficult to categorise.

The mean (SD) of reported activity problems was 6.1 (3.2). The median performance score was 4.8 (3.5; 6) and the median satisfaction score was 3 (2; 4.5). Activity problems within the categories productive and leisure activities were reported by 97% and 93% of the patients, respectively, whereas activity problems within the self-care category were reported by 53% of the patients (Figure 2). The subcategory reported by most patients was paid/unpaid work, which was reported by 84% patients. This was followed by household management (73%), socialisation (73%) and active recreation (67%) (Figure 3).

The frequency of activity problems distributed among the overall categories did not change remarkably from before to after prioritising (Figure 4 and Figure 5). However, among the subcategories, the distribution changed. Before prioritising, the subcategories household management (23%), paid/unpaid work (21%) and active recreation (17%) accounted for the most activity problems (Figure 6). After prioritising, paid/unpaid work was the largest subcategory, accounting for 27% of all prioritised activity problems, followed by socialisation (18%) and active recreation (17%) (Figure 7).

### 3.4. Post-COVID-19 Functional Status Scale

Current functional status evaluated with the Post-COVID-19 Functional Status Scale was completed by 446 (99.6%) patients (Table 3). Among these patients, 427 (98%) also rated their pre-COVID-19 functional status with the Post-COVID-19 Functional Status Scale. The reason for missing values is oversight by the occupational therapist assessing the patient. In general, the patients reported a lower level of functional status at assessment compared with the pre-COVID-19 infection period (2 (1) vs. 0 (0)). Moreover, 94% of the patients were categorised with no to negligible functional limitations before receiving a COVID-19 diagnosis. This number had decreased to 5% by the time of the assessment. In contrast, 94% patients reported slight to moderate functional limitations at their assessment.

## 4. Discussion

The aim of this study was to describe patient-reported mental fatigue, ADL problems, ADL ability, sick leave and functional status among patients with long COVID. At the time of assessment, 75% of the patients were categorised with moderate to severe mental fatigue, and the majority reported problems performing ADL. The median performance and satisfaction scores of the prioritised activity problems were 4.8 (3.5; 6) and 3 (2; 4.5), respectively. Collectively, the results suggest that the vast majority of patients had difficulties performing activities within the fields of productivity and leisure. At the assessment, 56% of the patients were on sick leave, and 94% of the patients reported a decreased functional status (Post-COVID-19 Functional Status Scale grade 2 or higher). In total, 94% of the patients had a rehabilitation need and were referred to rehabilitation in their home municipality.

We found a proportion of 73% females in our study population, which is larger than the proportion of females with a long COVID diagnosis with contact to a hospital in the Central Denmark Region (67%) [37]. In a systematic review and meta-analysis, female gender was found associated with a higher risk of developing fatigue (odds ratio = 1.54) [38], which may explain the difference above. However, their study was based on hospitalised patients, and since the majority of the participants in the present cohort were not hospitalised (86%), no direct comparison between the two studies can be made. Thus, the gender differences in relation to the impact of long COVID on activity performance, mental fatigue and work ability require further investigation in future studies.

With 51% of the patients having completed a medium- or long-cycle higher education, the share of comparatively well-educated individuals in the present study is considerably higher than the corresponding share in the general Danish population [39]. We do not know why this is so, but speculate that patients with a higher educational level have more demanding job positions and tasks requiring high cognitive and mental functioning. With mental fatigue, these functions are highly affected, and these patients may, therefore, experience a greater impact from long COVID and be more prone to seek help. Furthermore, several studies have found that people of a lower socioeconomic status carry a higher risk of facing social inequality in health, and the patients with lower educational level may face barriers for seeking treatment for their long COVID [40,41]. Thus, in future studies, it seems relevant to explore the possibility of social inequality in health among patients with long COVID.

By the time of their assessment, 56% of the patients were on sick leave. Among these patients, 96% were either on full-time or part-time sick leave due to their long COVID. The mean time of 267 days from the acute phase of COVID-19 to the assessment, corresponding to almost 9 months, indicates that more than half of the patients were on long-term sick leave. As long-term absence from work is a risk factor for exclusion from the work force [42]. Moreover, there is a potential risk that patients with long COVID may be marginalised and excluded from the labour market unless they are secured an effective and focused rehabilitation intervention aiming to improve their return to everyday life and work.

The reason for the long-term sick leave is uncertain, but studies have found that fatigue is associated with reduced ability to return to work and with decreased work status in the workplace among persons with acquired brain injury [43,44]. Since 94% of the patients were categorised with any level of mental fatigue in this cohort, this may explain the high proportion of patients on sick leave. However, further research exploring long-term sick leave and the association with mental fatigue in this patient group is warranted.

Since the Mental Fatigue Scale has not been validated for this group and was developed mainly for patients with neurological disorders, the instrument may be biased towards not reflecting the patients’ actual level of mental fatigue. However, the Mental Fatigue Scale includes many aspects described as typical long COVID symptoms and it has shown to potentially provide researchers and clinicians with deeper insight into mental fatigue within this patient group. Moreover, 38% of the patients were categorised with severe mental fatigue (Mental Fatigue Scale score ≥ 20), which is similar to findings from a recent study conducted by Davis et al. (2021). They characterised long COVID among an international cohort of 3762 persons with confirmed or suspected COVID-19. They assessed fatigue using the Fatigue Assessment Scale and found that 41% of unrecovered patients were experiencing extreme levels of fatigue compared with 9% of recovered patients [9]. They also found that unrecovered patients generally had higher Fatigue Assessment Scale scores than recovered patients and that memory issues and cognitive dysfunction influenced ADL and work ability. Therefore, the results of the Mental Fatigue Scale are likely to reflect the patients’ actual level of mental fatigue, but an evaluation of the instrument’s validity and reliability in this patient group is recommended.

Our results confirmed previous findings and shed novel light on patient-reported activity problems and their impact on everyday life among patients with long COVID. Due to the complexity and heterogeneity of long COVID, the use of the COPM secures a more comprehensive insight into the impact of long COVID on ADL than reported in earlier studies [9,18,19,20,21]. In a study evaluating a rehabilitation programme for patients with long COVID, Vanichkachorn et al. (2021) found that more than a third of the patients had difficulties performing ADL and 84% had difficulties performing household chores and work tasks [18]. However, they did not study ADL in subcategories. Thus, by using the COPM to look at ADL in subcategories, our study shines novel light on hitherto unexplored aspects of ADL problems and ADL ability following long COVID; aspects that should be studied further in the future.

Furthermore, the change in the distribution of activity problems within the subcategories from before to after prioritising suggests that aspects of paid/unpaid work, socialisation and active recreation are important to the patients and should, therefore, be taken into consideration when developing and optimising rehabilitation interventions for this patient group.

When categorising the activity problems identified in the COPM, we left out 60 activity problems reported by 32 patients due to difficulties categorising them based on the formulations or because they were not at the level of activity or participation. Although leaving out some of the reported activity problems might have caused an underestimation of activity problems per se, it is considered a strength of the study as this meant that we considered only activity problems at the activity and occupation level of the Taxonomic Code for Occupational Performance. This is considered to increase the content validity of the COPM [32].

The results of the Post-COVID-19 Functional Status Scale showed a significant and clinically relevant change in median score (0 vs. 2) between pre-COVID-19 infection and assessment at the clinic. At the time of their assessment, 59% of the patients were identified with slight functional limitations. However, whether a “slight functional limitations” score really reflects the patients’ functional status is debatable as 56% of the study population was on sick leave and 94% had a rehabilitation need due to their long COVID diagnosis. Hence, the patients’ functional status may potentially be more negatively affected than suggested by the Post-COVID-19 Functional Status Scale scores. This may potentially be explained by barriers to implementation of the instrument in the clinic. The instrument was developed in the early phase of the pandemic by Klok et al. (2020) and was shortly after translated into Danish. The manual of the Danish version differs from the English manual as the interview guide is not included. Moreover, only the construct validity of the English version has been examined [34]. Thus, the validation and reliability of the Danish version have yet to be examined. Additionally, the measurement of a pre-COVID-19 Post-COVID-19 Functional Status Scale score may potentially be affected by recall bias. However, since the Post-COVID-19 Functional Status Scale concerns overall functional status and general functioning in everyday life, the risk of recall bias is considered minimal.

### Limitations

The study had some limitations. The study was a single-centre study, which can be considered a limitation. However, the large sample size, a geographically large inclusion area covering both urban and rural districts, inclusion of patients during a long study period and a high completion rate of the outcome measures increase the study’s strength.

The included patients need to be considered as a selected population given the pattern of referral, where the patients’ first contact is their general practitioner who refer them to the outpatient Post-COVID-19 Clinic and from there to the occupational therapy assessment. As such, the study must be considered a sample study with all the built-in caveats that come with it, including generalisability. Moreover, we only included patients referred to occupational therapy and not patients referred to physiotherapy from the Post-COVID-19 Clinic. This may potentially influence the generalisability of the findings, and we consider this a limitation of the study. However, the patients referred to occupational therapy accounted for 93% of all patients assessed in the Post-COVID-19 Clinic, and of the 477 patients eligible for inclusion, 94% accepted participation in the study. Therefore, the generalisability of the findings remains substantial. Additionally, the study population only consisted of adult patients, as only patients ≥18 years old were referred to the Post-COVID Clinic following the national referral criteria. Although studies suggest that children also experience long COVID, albeit at a lower rate compared to adults, evidence is still sparse [45,46]. Thus, to ensure that all age groups are represented within the research on long COVID and its impact on everyday life, future studies are recommended to investigate everyday life and functioning among children with long COVID.

Indeed, in this new group of patients emerging in the wake of the COVID-19 pandemic, it was impossible to use any validated instruments. Due to the urgency of research on this topic, the importance of commencing the study before having a validated instrument was thought to outweigh the problems of potential biases.

Our study suggested that everyday life of patients with long COVID is highly influenced, but since our aim was to describe patient-reported mental fatigue, ADL ability, ADL problems, work ability and functional status, we did not study causalities. Hence, future studies are recommended to look at causalities between the included outcomes and long COVID.

## 5. Conclusions

Our study showed that the everyday life was highly influenced among the patients with long COVID assessed by occupational therapists at a Danish Post-COVID-19 Clinic. The majority were suffering from moderate to severe mental fatigue, and the patients’ ADL ability and activities of daily living were vastly affected. More than half were on sick leave by the time of assessment, and our results suggest a decrease in functional status from before infection with SARS-CoV-2 virus to assessment by the occupational therapist.

Our study contributes to and extends current knowledge on the impact of long COVID on everyday life, which is valuable for both researchers and clinicians when evaluating and developing rehabilitation interventions for patients with long COVID. However, as our study is a cross-sectional study, further research on the causalities of symptoms and their impact on everyday life is highly needed.

## Figures and Tables

**Figure 1 ijerph-19-14739-f001:**
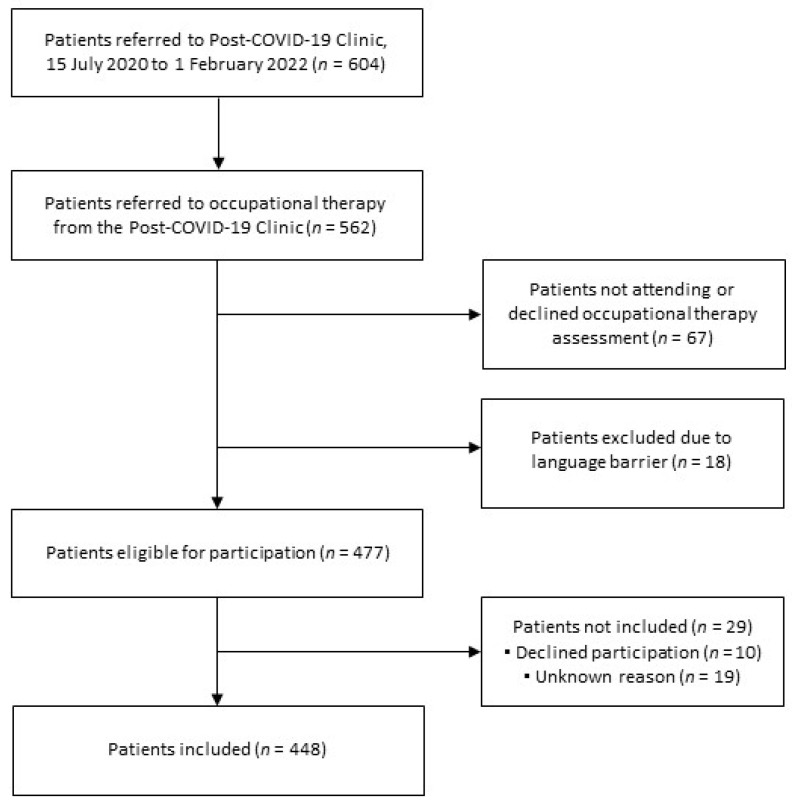
Flowchart of the participants showing the inclusion and exclusion flow.

**Figure 2 ijerph-19-14739-f002:**
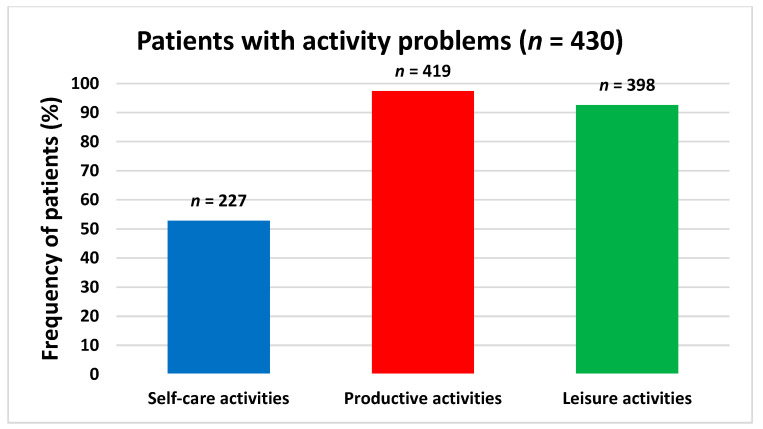
Patients with activity problems within the overall COPM categories: self-care activities (blue), productive activities (red) and leisure activities (green). Frequency (*n*) and relative frequency (%) of patients reported. COPM data is based on 430 patients.

**Figure 3 ijerph-19-14739-f003:**
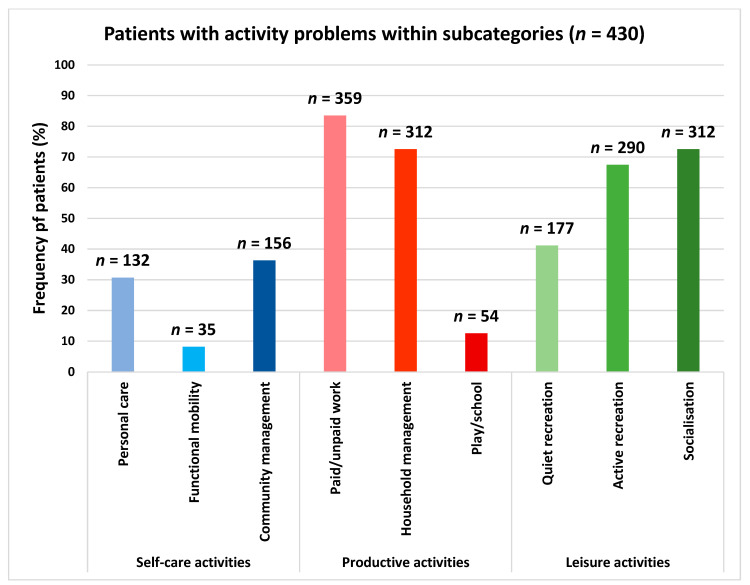
Patients with activity problems within the COPM subcategories: self-care (blue) comprising personal care, functional mobility and community management; productive activities (red) comprising paid/unpaid work, household management and play/school; leisure activities (green) comprising quiet recreation, active recreation and socialisation. Frequency (*n*) and relative frequency (%) of patients reported. COPM data based on 430 patients.

**Figure 4 ijerph-19-14739-f004:**
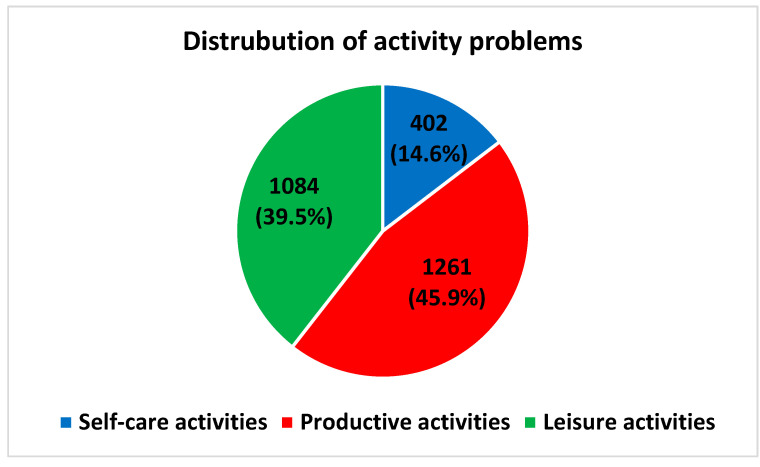
Distribution of reported activity problems within overall COPM categories: self-care activities (blue), productive activities (red) and leisure activities (green). Frequency (*n*) and relative frequency (%) reported. COPM data is based on 430 patients.

**Figure 5 ijerph-19-14739-f005:**
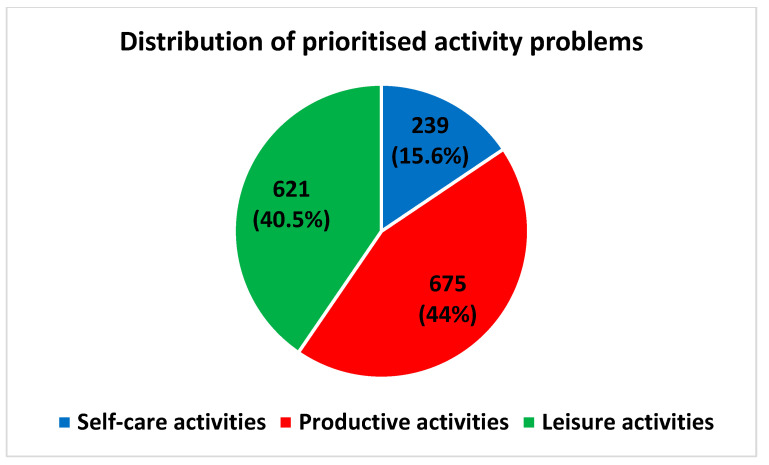
Distribution of prioritised activity problems within overall COPM categories: self-care activities (blue), productive activities (red) and leisure activities (green). Frequency (*n*) and relative frequency (%) of patients reported. Patients were the opportunity to prioritise up to five activity problems (see Materials and Methods). COPM data is based on 415 patients.

**Figure 6 ijerph-19-14739-f006:**
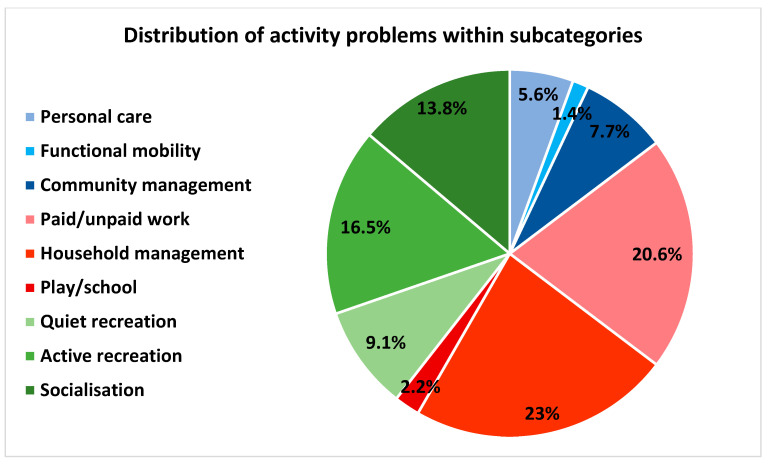
Distribution of activity problems (%) within the COPM subcategories: self-care (blue) comprising personal care, functional mobility and community management; productive activities (red) comprising paid/unpaid work, household management and play/school; leisure activities (green) comprising quiet recreation, active recreation and socialisation. COPM data is based on 430 patients.

**Figure 7 ijerph-19-14739-f007:**
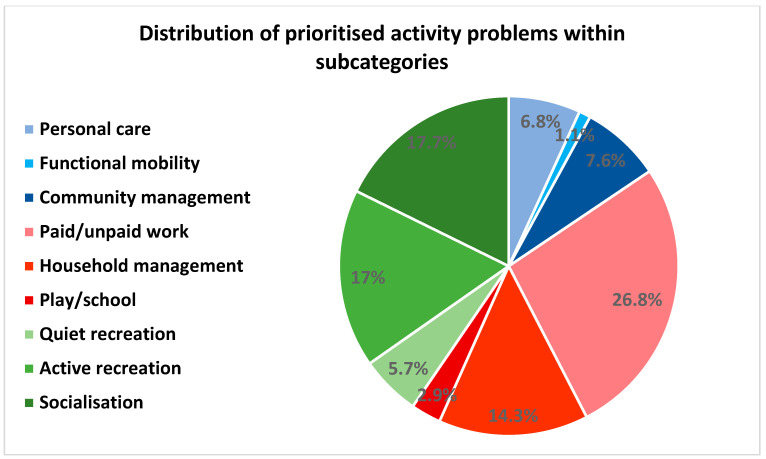
Distribution of prioritised activity problems (%) within the COPM subcategories: self-care (blue) comprising personal care, functional mobility and community management; productive activities (red) comprising paid/unpaid work, household management and play/school; leisure activities (green) comprising quiet recreation, active recreation and socialisation. Patients were the opportunity to prioritise up to five activity problems (see Section 2). COPM data is based on 415 patients.

**Table 1 ijerph-19-14739-t001:** Participant characteristics and sociodemographic data.

Sociodemographic Data		*n*
**Gender (females/men), *n* (%)**	325 (72.5%)/123 (27.5%)	448
**Age (y), mean (SD)**	46.8 (12.6)	448
**Living with spouse/partner (yes), *n* (%)**	333 (74.3%)	448
**Children living at home (yes), *n* (%)**	213 (47.5%)	448
**Children under 10 years old (yes), *n* (%)**	94 (44.13%)	213
**Highest level of education, *n* (%)**		446
No completed education, primary and lower secondary school	34 (7.6%)	
Upper secondary school	44 (9.9%)	
Short-cycle higher education	140 (31.4%)	
Medium-cycle higher education	170 (38.1%)	
Long-cycle higher education	58 (13.0%)	
**Work status before COVID-19, *n* (%)**		448
Employed/enrolled in education	401 (89.5%)	
Unemployed	47 (10.5%)	
**Current work status, *n* (%)**		401
Working/studying same hours as before	158 (39.4%)	
Sick leave	225 (56.1%)	
Unemployed	18 (4.5%)	
**Sick leave—reason and type, *n* (%)**		224
Part-time sick leave due to COVID-19	131 (58.5%)	
Full-time sick leave due to COVID-19	84 (37.5%)	
On sick leave for other reasons	9 (4%)	
**Time since infection (days), median (25th; 75th percentiles)**	246 (167.5; 346)	448
**COVID-19 test, *n* (%)**		448
Tested positive (PCR/antigen)	408 (91.07%)	
Positive antibody test	25 (5.58%)	
No test	15 (3.35%)	
**Hospitalised due to COVID-19 (yes), *n* (%)**	63 (14.1%)	448
Respirator (yes), *n* (%)	<4	63
**Patient-reported worst symptom, *n* (%)**		414
Respiratory issues	51 (12.3%)	
Other physical issues (such as joint pain, muscle weakness)	37 (8.9%)	
Mental fatigue and other cognitive deficits	293 (70.8%)	
Changed or reduced taste and smell	13 (3.1%)	
Other	20 (4.8%)	
**Referred to rehabilitation (yes), *n* (%)**	421 (93.8%)	448

**Table 2 ijerph-19-14739-t002:** Mental Fatigue Scale. Mean score, degree of mental fatigue and 24 h symptom variations. The scores reflect the patients’ current conditions (the past month) with long COVID compared to before infection with SARS-CoV-2 virus.

Mental Fatigue Scale		*n*
**Mental Fatigue Scale score, mean (SD)**	18.6 (5.6)	447
**Mental Fatigue Scale categories, *n* (%)**		447
No mental fatigue	27 (6.0%)	
Slight mental fatigue	85 (19.0%)	
Moderate mental fatigue	164 (36.7%)	
Severe mental fatigue	171 (38.3%)	
**24-hour variations, *n* (%)**		444
Have noticed no difference during the day and night	83 (18.7%)	
Clear difference between certain times of the day	279 (62.8%)	
Feeling unwell all times, day and night	82 (18.5%)	
**Feeling best, *n* (%)**		250
Morning	190 (76%)	
Afternoon	24 (9.6%)	
Evening	32 (12.8%)	
Night	4 (1.6%)	
**Feeling worst, *n* (%)**		215
Morning	32 (14.9%)	
Afternoon	102 (47.4%)	
Evening	70 (32.6%)	
Night	11 (5.1%)	

**Table 3 ijerph-19-14739-t003:** The Post-COVID-19 Functional Status Scale. Evaluated in collaboration with the occupational therapist during assessment. Post-COVID-19 Functional Status scores at assessment are based on the current long COVID symptoms and general condition.

Post-COVID-19 Functional Status Scale	Pre-COVID-19 Diagnosis (*n* = 437)	At Assessment (*n* = 446)
Median (25th; 75th percentiles)	0 (0; 0)	2 (2; 3)
0: No functional limitations	339 (77.6%)	3 (0.7%)
1: Negligible functional limitations	71 (16.3%)	17 (3.8%)
2: Slight functional limitations	22 (5.0%)	261 (58.5%)
3: Moderate functional limitations	<5 (<1%)	157 (35.2%)
4: Severe functional limitations	<5 (<1%)	8 (1.8%)

## Data Availability

The data sets are not publicly available due to signed consent agreements around data sharing, but the authors will look into possibilities for data sharing on reasonable requests.

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
