# Peer review of "Mental Fatigue, Activities of Daily Living, Sick Leave and Functional Status among Patients with Long COVID: A Cross-Sectional Study"

_ijerph, 2022, doi:10.3390/ijerph192214739_

Round 1
Reviewer 1 Report
ID: ijerph-1969010
Title: Long COVID – Impact on Everyday Life
Thank you for providing a chance to review this manuscript.
Comment: major revision.
Detailed information:
Title
Please be more specific and clearer.
Introduction
Line 37, page 1: What is the theoretical basis for the sentence “The majority of people infected with SARS-CoV-2 virus causing COVID-19 recover without treatment and return to normal life”?
Line 42-52, page 1-2: There are problems with the language organization and logic of this paragraph. Please correct and simplify the sentence appropriately.
Line 69, page 2: The last paragraph of the introduction is often the part that readers are more interested in. But your description of the hypothesis is not clear, and
you should think well about this part and make reasonable changes.
Materials and Methods
Line 71, page 2: Where did the subjects come from? What is the sample size? What are you basing this on? It can be seen from the results that there are many missing values in your data, how to carry out quality control?
Line 81-83, page 2: I don't think that's a good exclusion criterion.
Line 102-140, page 3: What is the reliability and validity of all scales? How do you ensure the scale is scientific?
Line 142-147, page 3: Descriptive statistics only? The objectives of the study were that evaluate how long COVID affects patient-reported mental fatigue, ADL problems and performance, sick leave and functional status. The results show the measurement of each scale, then how to explain these results are related to the long COVID?
Results
Table 1-3, page 5-10: Using a three-line table will make your results more beautiful and clearer.
Figure 5 and 7, page 8 and 10: As mentioned above, "In total, 430 patients completed the COPM, but 15 patients did not prioritize their 196 activity problems "contradicts the total number of people 425 in the figure. I think you should check these unnecessary mistakes carefully.
Table 3: Does this table have missing values?
Overall: Subheadings can be used for the results section according to the results of the scale measurement. This will make it clearer to the reader. In addition, the results presented in the COMP section need to be more logical and clearer.
Discussion
Line 253-268, page 10: This paragraph is lengthy and unnecessary. Please simplify or delete it.
Overall: 1) The discussion is not to retell the results, but to explain the reasons and to demonstrate with the evidence of previous research. Please delete unnecessary words and reorganize your structure. 2) Please discuss in the order in which your results are presented. 3)Please re-segment and merge the main idea paragraphs, which can make your logic clearer.
Conclusion
Line 378-384, page 12: Here is still a summary of your results, please make a targeted conclusion. Please simplify your conclusions.
Firstly, reading more articles from the TOP health quality journals, to learn the formats, expressions, and of great importance—logic, might help a lot before revising. Secondly, please avoid low-level errors in your results and use three-line tables to make the format more formal. Finally, as it was written in the discussion, I think there are many problems with your research, the biggest of which is a lack of innovation. Please show more new findings in this article in more detail compared to other articles.
Thank you and my best,
Your reviewer
Reviewer 2 Report
The authors take up the very current subject of the chronic consequences of post-Covid-19.
Despite the value of the data provided, some aspects need to be improved.
The following are my suggestions and comments to the authors:
1. Maintain structure abstract:: background, the aim of the study, methodology, results, conclusions
Adds statistical significance when describing the results.
I2. n section 2. Materials and Methods
-line 114 changes the title of the subsection should not be shortcut 2.2. COPM
-add information on how you assessed sick leave
- line 141, in the description section 2.4. Statistical analysis
add information about what statistical tests you used, what significance level was used, what power of the test was used
-on line 153 you write "Written consent was obtained from each patient ahead of any study procedures/"which procedure did you follow? you write that this is an observational study; please explain
3. In the results section
- in table 1, add information about the number of men,
additionally present the results of the statistical analysis, enter the p value for the compared results within the variables,
-in table 2 also provide the p value for the analyzed variables
--add a description of the colors under the Figure 1 and 2
- in table 3 compare the groups, enter the value of p
4, In section „ Discussion” I propose to add the information that the Covid-19 pandemic caused a decrease in the attendance of health care workers and that knowledge about the pandemic in some medical groups increased absenteeism. I suggest to add to literature :
doi: 10.3390/healthcare9040398. PMID: 33916092.
5, Add a Limitation section
6. Modify the Conclusions section -
present the conclusions of the research carried out,
do not include your results in the conclusions.
Reviewer 3 Report
Dear author
I'm glad to see hands-on engagements about the health of long-term COVID-19 patients. This is an important thing to do to end this wave and extend the executive decision to find follow-up medical care. However, manuscript visualization is important and the following suggestions will be made for improvement.
1. Abstract
Lines 28-30, please clearly point out the problems of the new crown patients, what needs to be improved?
2.Introduction
Lines 68-69, I hope the authors can summarize the results of these literatures exploring the symptoms and occupational impact of COVID-19.
3.Materials
This is a rigorous manuscript. It is believed that the author cites many well-known questionnaire samples to conduct the survey. However, not every questionnaire design can be seamlessly integrated and suitable for follow-up research subjects. Therefore, it is suggested that the authors supplement or add questionnaire reliability and validity analysis to strengthen the applicability of the used questionnaire to this manuscript.
Supplement the research architecture diagram, which makes it easier for the reader to understand the focus of this manuscript.
4. Results
Analyzing data is simple, but focused, which is a good thing.
5.Discussion
Although the discussion is based on the analysis results to illustrate. But ask, what is the result of inverse each analysis?
I think the discussion is about adjusting your elbows based on your experience and opinions. to get meaningful results.
good luck ,
Round 2
Reviewer 1 Report
ID: ijerph-1969010
Title: Mental fatigue, activities of daily living, sick leave and functional status among patients with long COVID: A cross-sectional study
Thank you for providing a chance to review this manuscript.
Comment: minor revision.
Detailed information:
Title
The title you wrote in the two documents is different. Please pay attention to this detail. The title in “ijerph-1969010-peer-review-v2” should be change to “Mental fatigue, activities of daily living, sick leave and functional status among patients with long COVID: A cross-sectional study” in “ijerph-1969010-coverletter”.
Materials and Methods
As you said, you chose a cross-sectional study. Is it a census or a sample survey?Why not consider the sample size? The sample size is affected by other factors such as expected prevalence, admissible error and required significance level.
More thought should be given to the significance of the article.
Thank you and my best,
Your reviewer
Author Response
Title:
Thank you for observing the difference in the titles in the two documents. The title in the cover letter has now been changed and is identical to the revised title of the manuscript.
Materials and Methods:
We thank the reviewer for the opportunity to clarify our thoughts.
First, a comment to choice of survey.
We are aware that the included patients need to be considered as a selected population following the pattern of the referral (General Practitioner – Outpatient Post-COVID-19 Clinic – Occupational Therapy). Thus, we have addressed and adjusted the limitation section accordingly, please see lines 404-408, p. 13.
Second, a comment to sample size.
Since knowledge on the true prevalence of the reported outcomes of the present study is very limited and fraught with considerable uncertainty, we applied a descriptive statistical approach not based on sample size calculation, which presupposes an estimated occurrence of the investigated outcomes.
We hope that the clarification of our thoughts above and the changes made in the limitation section accommodate reviewer's comment to the Materials and Methods section.
Reviewer 3 Report
Dear authors,
I am very happy to receive the revised manuscript. This manuscript has further additions than the previous version. I think that the main issues and details of the current manuscript have been greatly improved, and I believe that readers can understand this research report more clearly. I think it is currently at the level of publication and recommend that the editor consider accepting this manuscript.
good luck,
Author Response
We thank reviewer for the comment and the recommendation of publication to the editor.